# Pharmacokinetic Basis for Using Saliva Matrine Concentrations as a Clinical Compliance Monitoring in Antitumor B Chemoprevention Trials in Humans

**DOI:** 10.3390/cancers15010089

**Published:** 2022-12-23

**Authors:** Dinh Bui, Lenora A. McWilliams, Lei Wu, Haiying Zhou, Stuart J. Wong, Ming You, Diana S.-L. Chow, Rashim Singh, Ming Hu

**Affiliations:** 1College of Pharmacy, University of Houston, Houston, TX 77204, USA; 2College of Nursing, University of Houston, Houston, TX 77204, USA; 3Simulations Plus Inc., Lancaster, CA 93534, USA; 4Department of Neoplastic Diseases, Medical College of Wisconsin, Milwaukee, WI 53226, USA; 5Houston Methodist, Houston, TX 77030, USA

**Keywords:** cancer chemoprevention, OCT transporter, PBPK modeling, plasma-saliva correlation, entero-salivary recycling, pharmacokinetics, saliva excretion marker, patient compliance tracker

## Abstract

**Simple Summary:**

This is the first human study using saliva samples for drug therapeutic monitoring and patient compliance for a complex botanical extract (e.g., ATB) and reporting the rapid and extensive secretion of certain active compounds (matrine, dictamnine) to human saliva. Specifically, matrine is among the compounds with the highest excretion ratio to saliva with the ratio of saliva/plasma of 6.4 ± 1.4 for C_max_ and 4.8 ± 1.7 for AUC in healthy adults. The results were further analyzed through a population PK model and a PBPK model, and the compound tracer (e.g., Matr) could serve as a competitive biomarker that enable further development of the salivary secretion based PK/PD analysis. Matrine has demonstrated to be a promising compound to study the drug transport to saliva and a marker compound to follow patient compliance.

**Abstract:**

This study reports the first clinical evidence of significantly high secretion of matrine in a multi-component botanical (Antitumor B, ATB) into human saliva from the systemic circulation. This is of high clinical significance as matrine can be used as a monitoring tool during longitudinal clinical studies to overcome the key limitation of poor patient compliance often reported in cancer chemoprevention trials. Both matrine and dictamine were detected in the saliva and plasma samples but only matrine was quantifiable after the oral administration of ATB tablets (2400 mg) in 8 healthy volunteers. A significantly high saliva/plasma ratios for C_max_ (6.5 ± 2.0) and AUC_0–24_ (4.8 ± 2.0) of matrine suggested an active secretion in saliva probably due to entero-salivary recycling as evident from the long half-lives (t_1/2_ plasma = 10.0 ± 2.8 h, t_1/2_ saliva = 13.4 ± 6.9 h). The correlation between saliva and plasma levels of matrine was established using a population compartmental pharmacokinetic co-model. Moreover, a species-relevant PBPK model was developed to adequately describe the pharmacokinetic profiles of matrine in mouse, rat, and human. In conclusion, matrine saliva concentrations can be used as an excellent marker compound for mechanistic studies of active secretion of drugs from plasma to saliva as well as monitor the patient’s compliance to the treatment regimen in upcoming clinical trials of ATB.

## 1. Introduction

Antitumor B (ATB) or Zeng Sheng Ping is a botanical drug, consisting of water extract or powder of six plants: *Sophora tonkinensis* Gagnep., *Polygonum bistorta* L., *Prunella vulgaris* L., *Sonchus brachyotus* DC., *Dictamnus dasycarpus* Turcz., and *Dioscorea bulbifera* L (Table 1). The anticancer effect of ATB has been demonstrated in mouse models of bladder cancer [1], lung cancer [2,3], and oral cancer [4]. Our group successfully identified several key active compounds (KACs) of ATB in the A/J mouse oral tissues that are capable of inhibiting oral cancer cell proliferation and likely contribute to the chemoprevention effects of ATB [3]. The four major KACs responsible for the anticancer activities of ATB were matrine (Matr), dictamine (Dict), fraxinellone (Frax), and maackiain (Maac) [3]. ATB has also shown significant chemopreventive efficacy against human esophageal and oral cancers in several clinical studies in China [5,6,7,8]. ATB is currently undergoing clinical trials (ClinicalTrials.gov Identifier: NCT04278989) in cancer patients for chemopreventive efficacy against oral cancer in the United States.

In chemoprevention clinical trials, where trial duration is usually longer (1–5 years or more), ensuring patient compliance is essential for a successful study design and execution. Therefore, patient monitoring for adherence to the medication regimen without using invasive procedures would significantly improve trial integrity and data quality of chemopreventive clinical studies. Our animal studies have shown that matrine is secreted via saliva in mice after i.p. administration of ATB [9]. Therefore, we hypothesized that matrine secreted in human saliva can be used for compliance monitoring and act as a possible indicator of drug exposure at the target site. There has been no report published so far that examines the secretion of matrine in human saliva and very limited information about the human pharmacokinetics (PK) of ATB herbal mixture is available. Matrine is of particular importance as it is also reported to have anticancer activities [10,11] and is used as a quality control marker of ATB as per the Chinese Pharmacopeia.

Therefore, the purpose of the present study is to advance the understanding of ATB pharmacokinetic behaviors in preparation for oral cancer chemoprevention trials in humans. In the present study, we aim to: (1) Characterize the pharmacokinetics of ATB-KACs in healthy humans; (2) Conduct interspecies scaling of the PK parameters from mice to estimate the PK profiles of ATB-KACs in rats and humans; and (3) Model the correlation between drug concentrations in plasma and saliva such that saliva concentrations of active compounds can be successfully used to represent their plasma and oral cavity exposure, and monitor patient compliance in the upcoming and future clinical trials.

## 2. Materials and Methods

### 2.1. Materials

ATB (300 mg tablets) manufactured according to Chinese GMP standards was purchased from Central Pharmaceutical Co. Ltd., Tianjin, China. Baohuoside I (Bao) was purchased from Chengdu Must Bio-technology Co. Ltd. (Chengdu, China). Ammonium acetate (LC-MS grade) was procured from Sigma (St. Louis, MO, USA). Frax and Matr were purchased from TCI (Portland, OR, USA), whereas Dict and Maac were purchased from Ambeed (Arlington Heights, IL, USA). Saliva sample collection kits were purchased from Salimetrics (State College, PA, USA). All other materials (typically analytical grade or better) were used as received.

### 2.2. Stability of ATB Tablets

Two independent batches of ATB tablets were tested according to International Conference on Harmonization (ICH) Guideline for long-term (12 months at 25 ± 2 °C/60 ± 5% RH), intermediate (6 months at 30 ± 2 °C/65 ± 5% RH), and accelerated (40 ± 2 °C/75 ± 5% RH) stability [12]. Each ATB tablet weighed 330 mg, and contained 300 mg ATB extract equivalent to 296.9 ± 56.2 μg Matr, 3.9 ± 0.8 μg Dict, 5.1 ± 0.2 μg Maac and 7.0 ± 1.3 μg Frax (Table 2).

### 2.3. Quantitative Analysis

A validated UPLC-MS/MS to analyze ATB KACs (Matr, Dict, Maac, and Frax—Figure 1) was developed as per FDA guidance [13] and as published before [14]. The complete details of method development, validation, and sample preparation were included in Appendix A. Samples were analyzed in Waters Acquity™ and AB Sciex 5500 mass spectrometer (Applied Biosystem/MDS SCIEX, Foster City, CA, USA) equipped with an APCI TurboIonSpray™ source using compound dependent parameters listed in Appendix A. The chromatogram of ATB KACs and the internal standard (IS) were shown in Figure 2.

### 2.4. Pharmacokinetics of ATB Compounds in Healthy Adults

An open-labeled, single-dose oral pharmacokinetic study (NCT04230057) of ATB (8 tablets, Table 2) in healthy human subjects was conducted to determine the pharmacokinetic parameters of four ATB KACs in saliva and plasma. In addition, in vivo correlation between plasma and saliva concentrations was determined as the secondary outcome. The study supported the study design of the planned window of opportunity pharmacokinetic (NCT03459729) and efficacy (NCT04278989) clinical trials of ATB in oral cancer patients.

#### Study Design

Details of the study design and participant inclusion/exclusion criteria were reported in Appendix A. Healthy men and women (age between 18–40 yrs) with no diagnosed disease or illness were enrolled in the study. Briefly, each participant (fasted for 8–12 h) received single-dose oral administration of 8 tablets of ATB with 250 mL water. Blood (approximately 2 mL) and saliva (approximately 2 mL) samples were collected at pre-determined time points (0, 0.5, 1, 2, 3, 4, 6, 8, and 24 h). Saliva samples were collected by spitting into the Salimetric collection kit. Volunteers were asked to stop drinking or eating 15 min before sampling and spit into the saliva sample kit for salivary collection to reduce the variability in the volume of saliva sample collection. Blood samples were centrifugation at 4 °C, 3200× *g* for 15 min to obtain plasma samples. Breakfast was provided at 30 min and lunch at 3 h post-administration. Water and juice were permitted as desired except within 30 min of administration. Participants avoided any drink 15 min before each saliva sample. To mimic the future clinical trial in patients, no further standardization of saliva samples was made in the current study. Blood biochemistry labs were performed for each participant before enrollment in the study and 24 h after the administration of ATB. Safety was assessed throughout the study by observing the participants and post-study participant-reported adverse events, and blood chemistry laboratory assessments. A list of tests performed pre-enrollment and post-study are reported in Appendix A and blood sample biochemistry report is shown in Appendix A. The sample size in this study (8 participants) was a representative of other first in human exploratory tolerability, pharmacokinetic, and pharmacodynamic biomarker studies and based on the power calculation (alpha = 0.05, power = 80%) reported inter-individual variance of clearance of matrine in patients (60–83%, average 70%).

### 2.5. Pharmacokinetic Analysis and Modeling

The PK parameters (C_max_, T_max_, AUC, T_1/2_) of ATB KACs were calculated by the non-compartmental method, using WinNonlin 8.2 (Pharsight Corporation, Mountain View, CA, USA). The ratio of drug concentration in plasma and saliva were calculated using Equations (1) and (2).
(1)$$R_{C_{\max}}=\frac{C_{\max}\;saliva}{C_{\max}\;plasma}$$
(2)$$R_{\mathrm{AUC}}=\frac{{AUC}_{0–24}\;saliva}{{AUC}_{0–24}\;plasma}$$

#### 2.5.1. Pharmacokinetics Modeling

The population PK model was developed using plasma and saliva concentrations of 8 healthy participants in the study. No input was made for the missing data points. Initial estimates of individual compartmental PK parameters were derived from non-compartmental analysis. Different compartmental PK models were tested to describe the plasma and saliva PK profiles of Matr. The population pharmacokinetic compartmental models using actual dosing and sampling times were built using Phoenix NLME 8.2. and goodness-of-fit was determined. More details of PK modeling are described in Appendix A.

#### 2.5.2. Physiologically Based Pharmacokinetic (PBPK) Modeling and Interspecies Scaling

The PBPK model of Matr in blood/plasma was developed using GastroPlus^®^ 9.8 software (Simulations Plus, Inc., Lancaster, CA, USA). All physiological parameters have parameterized *a priory* in the software. The blood/plasma ratio and solubility parameters of matrine were used as reported [10,11]. The LogD, pKa, and permeability parameters were used as predicted by the ADMET Predictor™ 10.2 (Simulations Plus) (Table 3). Due to significant differences in the bioavailability of Matr upon intraperitoneal and oral administration [9], the first pass effect was considered an important contributor to the oral pharmacokinetics of Matr. However, the specific enzyme(s) involved in the gut metabolism of Matr has not yet been determined. It was earlier reported that Matr was not metabolized by CYP or UGT enzymes [15].

The model development method was using the i.v. data from mouse and rat to validate the PBPK Kp calculation method. The PBPK model of Matr PKs was developed with the i.v. PK profile of matrine in mice [9], rats [15], and the p.o. data in humans from the current clinical study NCT04230057. The i.v. data in mice and rats are used to establish and validate of drug volume distribution method. The calculated volume distribution was also compared with the non-compartment analysis of the i.v. data. The model was then extrapolated to humans by adding the first pass effect in the intestine [9]. A human-slow transit ACAT model was used to account for the delay in drug release due to the herbal matrix in the tablets. The compound and physiological parameters used to build the models were listed in Table 3. The major clearance mechanisms of Matr are renal excretion [11]; hepatic and intestinal first pass metabolism [14]; and salivary clearance reported in this study. The parameters used to develop the PBPK model of matrine were listed in Table 3. The PBPK model was evaluated for the closeness of simulated and observed data (C_max_, T_max_, AUC). The model was also validated externally by simulating the plasma concentration with the dose of 4.8 mg Matr in American males and compare with the reported data by Gao et al. [11]. More details of interspecies scaling in the PBPK model are described in Appendix A.

### 2.6. Statistical Analysis

All the data were presented as means ± RSD (relative standard deviation), if not specified otherwise. Significance differences between plasma and saliva PK parameters obtained from NCA analysis with Phoenix WinNonlin were assessed by using unpaired Student’s *t*-test with GraphPad Prism 9.1. *p*-value of <0.05 was considered as statistically significant.

## 3. Results

### 3.1. Quality Control of ATB Products

Since the chemoprevention trials of ATB will run for several years, stability of KACs content in the ATB tablet was established in different storage conditions. The results showed that all KACs were stable under tested conditions and Matr content in ATB tablets was significantly higher than other KACs (Appendix A, Table 2).

### 3.2. Safety of ATB in Healthy Adults

Six healthy men and 4 healthy women were enrolled in the study. Two participants (one man and one woman) withdrew from the study before the administration of the drug. Study data from 8 participants were included in the safety and pharmacokinetic analysis sets. The demographic of the participants was 2 Caucasian (25%) and 6 Asian (75%). Table 4 summarizes the age, weight, height, and BMI average of the study participants. No major adverse events were reported by the participants within 24 h administration of ATB. Certain changes in the blood biochemistry of individual participants were observed but not deemed to be remarkable (Appendix A). No abnormal changes were observed in the liver function (AST, ALT, GGT) of any participant and no acute adverse effects were observed by the study team or reported by the participants within 24 h of the ATB administration.

### 3.3. ATB Pharmacokinetics in Human Plasma and Saliva

A sensitive and validated UPLC-MS/MS method with rapid analysis (run time 6.5 min), small samples volume requirement (20 µL), simple sample preparation, good recovery, and minor matrix effect for simultaneous quantification of ATB-KACs in human plasma and saliva was developed, as published before in rodent [9]. The validation results were shown in Appendix A. This method will be adopted for analyzing blood and saliva samples in upcoming clinical trials of ATB.

Determination of plasma and saliva concentrations of four KACs after 2400 mg ATB oral administration showed only Matr in quantifiable concentration in both plasma and saliva of all participants, which also corresponds to its highest content in the ATB tablet. Dict was detected below or close to the lower limit of quantification (7.9 nM) in one-hour plasma sample of one subject (09M) and 30 min saliva samples of two participants (01F and 09M). Maac was detected in 30 min plasma sample of only one participant (08F). Frax was not detected in any study sample. Even after concentrating the samples to enrich the KACs amounts, no other KAC except Matr could be quantified. Appendix A showed the plasma and saliva concentrations profile of Matr for individual study participants and Figure 3 showed the average of all 8 subjects. Table 5 summarizes the estimated PK parameters of each participant using non-compartmental analysis.

Surprisingly, significant higher concentrations of Matr were found in the participant’s saliva at every time point compared to the plasma concentrations (Figure 3). Matr plasma half-life (10.0 ± 2.8 h) and T_max_ (2.0 ± 0.6 h) were not significantly different from its saliva half-life (13.4 ± 6.9 h) and T_max_ (1.5 ± 0.9 h). However, the average AUC_0–24_ of Matr in saliva (1214.5 ± 398.7 ng * h/mL) was much higher than in plasma (266.2 ± 80.0 ng * h/mL) with high saliva/plasma ratios for both C_max_ (6.5 ± 2.0) and AUC_0–24_ (4.8 ± 2.0). The saliva/plasma ratio of Matr is among the highest ratios that have been reported in the literature [17,18,19,20]. The plasma AUC_0–24_ of Matr was comparable to previously reported values in a PhD thesis [11]. The inter-subject variability of systemic exposure (AUC) was found to be high (30.1% in plasma and 32.8% in saliva). There was a total of 8 participants in the study and data from all participants was included in the compartmental and PBPK modeling.

### 3.4. Compartmental Co-Modeling of Matrine in Plasma and Saliva

Figure 4 showed three different PK models (A, B, and C) developed for Matr PK profiles in plasma and saliva using non-linear PK parameters V_max_ and K_m_ to describe the transport of Matr from plasma to saliva. Model A has two compartments plasma and saliva, while a tissue compartment was added to Model B and Model C. A direct absorption to the saliva compartment was added in Model C (Figure 4) to account for the reabsorption of drug excreted in the saliva from the oral cavity and the intestine, thereby entering the recycling process (Figure 5). Though we expect the contribution of oral cavity recycling and entero-salivary recycling to be small, its addition significantly improved model fitting (compared to Model B). Figure 6 showed the model structure of the PBPK model using GastroPlus^®^ 9.8.

The goodness-of-fit plots (Figure 7) indicated that Model C most adequately described the drug concentration in plasma and saliva, while showing a good correlation between observed and predicted concentrations in saliva. The addition of direct absorption to the salivary compartment resulted in a better AIC value (Model A = 1326, Model B = 947, and Model C = 824) and lower standard errors of the parameter estimates (coefficient of variation <25%). (Figure 7). Model A though adequately described the plasma concentration profiles of Matr, showed poor correlations between observed and predicted saliva concentrations. Model B relatively showed better fitting with the observed data in plasma and saliva than Model A, but still underestimated the saliva concentration (CWRES vs. IVAR plot, Appendix A).

The best fit compartmental Model C (Figure 5 and Figure 7) developed made it possible to use saliva concentration to predict the drug concentration in plasma. The model was validated using the internal approach by running a bootstrap of 500 sampling times and comparing the final model parameters with the current model parameters. The values of the parameters were within the 25% confidence interval. This proved that the model parameters (Table 6) were robust.

### 3.5. PBPK Modeling

A few PBPK models have been developed for botanical drugs in humans, although PBPK models are widely used to estimate the population PK parameters of many preclinical compounds and prescription drugs [10,21,22,23]. Notably, a PBPK model of Matr was developed in rats after oral administration of pure matrine or crude compound in ATB [10]. However, the study data was only limited in rats and did not report the drug concentrations in saliva as well as oral cavity tissues. The current PBPK model (Figure 6) explains the distribution and clearance of matrine after oral administration of ATB in different species, and successfully predicts the PK parameters in humans based on the rodent’s data. A comparison of the predicted versus actual concentration-time profiles (Figure 8, Table 7) shows that the PBPK model was able to simulate the experimental data closely and perform the interspecies scaling of the PK profiles.

The PBPK model was externally validated by closely predicting the time course of Matr concentration in the human plasma after administering 1.2 g ATB (equal to 4.8 mg Matr) reported by Gao et al. [11] (Figure 9).

## 4. Discussion

The human PKs of ATB revealed that we can use the salivary secretion of Matr, one of the key active compounds of ATB at the site of action (i.e., oral cavity) as a measurement of matrine exposure and marker of patient compliance in long-term chemoprevention clinical studies of ATB. The PBPK modeling studies indicated that PBPK modeling can be developed to describe the PK behaviors of Matr across three different species.

This is the first human study using saliva samples for drug therapeutic monitoring and patient compliance for a complex botanical extract (e.g., ATB). This significantly improves the study robustness and quality of long-term chemoprevention clinical trials by monitoring patient adherence to a study protocol, which should positively impact the quality and reliability of the collected data [24,25]. Additional benefit of using Matr concentration in saliva for oral cancer trials comes from its anti-inflammatory and anti-proliferative activity in the oral cavity [26]. Most importantly this study has provided the needed exposure information at the site of action that allows for the complete monitoring of drug PK profile using saliva samples for the time between patient visits during the window of opportunity pharmacokinetic (NCT03459729) and efficacy (NCT04278989) clinical trials of ATBs. Lastly, saliva samples represent a non-invasive, patient-friendly self-sampling method, which may reduce cost and improve patient retention in the study.

The saliva and plasma concentration profiles of Matr were successfully co-modeled using Model C (Figure 5 and Figure 6) which included the active secretion of Matr into saliva, presumably via an OCT transporter expressed in salivary glands [27,28]. The model structure includes plasma, saliva, and tissue compartments. Matr was rapidly absorbed (K_a_ = 1.66 ± 0.27 h^−1^) from the gastrointestinal tract, whereas comparatively slower absorption was estimated from the oral cavity (K_as_ = 0.39 ± 0.14 h^−1^). However, later is expected to have minor contribution to overall drug absorption due to tablet dosage form in the current PK study.

After absorption, Matr mostly stayed in plasma as evident from a significantly large volume of distribution of the central compartment (V_p_ = 1.42 ± 0.08 L/kg) as compared to saliva (V_s_ = 0.02 ± 0.01 L/kg) and tissue (V_t_ = 0.62 ± 0.04 L/kg), indicating significant protein binding of Matr. The estimated vs. of Matr in saliva was reasonable based on the normal daily production of saliva (0.5–1.5 L) in humans [29]. This also explains the higher drug concentrations of Matr in saliva samples.

The model estimated a rapid secretion of Matr from tissue to saliva (K_ts_ = 23.42 ± 2.33 h^−1^), whereas uptake of Matr from plasma into the tissues (K_pt_ = 0.11 ± 0.02 h^−1^) was the rate limiting step. In a pilot study performed in our lab in F344 rats, intravenous infusion of Matr resulted in the salivary excretion of Matr within 15 min. Model-estimated faster rate of reabsorption of salivary-secreted Matr from the oral cavity into plasma (K_sp_ = 1.25 ± 0.28 h^−1^) was comparable to its absorption from the gut.

The estimated high K_m_ values (279.52 ± 5.73 mg/L equal to 1125.5 µM) of Matr transport from plasma to saliva via an OCT transporter were comparable to K_m_ values of metformin (saliva/plasma concentration ratio = 0.29–0.39) uptake by OCT1, OCT2 (majorly expressed in kidney) and OCT3 (majorly expressed in saliva) in the range of 285–3170 µM [30]. However, K_m_ values of Matr need to be verified further in OCT-expressed cells.

The involvement of active transporter was evident from a significantly high ratio of saliva to plasma concentration at each time point for all study participants (Appendix A). The saliva-plasma level correlation allowed for the estimation of population PK parameters and provide useful information that can be integrated with a sparse blood sampling approach to provide more detailed and complete population pharmacokinetic profiles. The saliva concentration profile can be used as a surrogate for oral cavity exposure in the PK/PD modeling during the clinical trials and can help inform dose adjustment decisions, which is often done in chemotherapeutic settings [31,32,33,34].

The PBPK model of Matr successfully explains the distribution and clearance of Matr after oral and i.v. administration of ATB in different species with a good simulation of observed data (Figure 8, Table 7), and closely predicting the PK parameters in humans based on the rodent’s data. The rodent PK and metabolic profiling of other KACs suggest high pre-systemic clearance in humans, which is consistent with low KACs exposure in blood and saliva after oral administration of ATB tablets. The salivary drug concentration profile was not included in the PBPK model because most of the excreted drug will be reabsorbed in the GI tract and the involvement of transporter that is needed more investigation. Overall, the PBPK model of Matr can be successfully applied to build a predictive human PK profile from rodent PK data, which can provide insights into calculating the required dose of ATB to achieve the desired exposure in the oral cavity. Through oral administration, matrine exhibits dual PK profiles which were further analyzed through a population model and a PBPK model, and the compound tracer (e.g., Matr) could serve as a competitive biomarker that enables further development of the salivary secretion-based PK/PD analysis.

Several limitations with the current study exist that should be acknowledged and kept in mind while interpreting its results. First, we missed a few data points at the initial time points due to technical difficulty in blood sampling. However, no significant impact on the quality of the model due to missing data points suggests that we can successfully use sparse sampling for PK parameter estimation in the clinical trial design. Second, the total sampling duration was not ideal (ideal last time point ~ 5 × half-life) for estimating the half-life of Matr, because Matr apparent half-life was longer than expected, probably due to the extended elimination phase resulting from probable entero-salivary recycling (Figure 5, see Appendix A for more discussion). Therefore, in future clinical trials, saliva samples will be collected daily for 14 days which will allow us to use co-modeling approach to determine the PK parameters and more accurately estimate the half-life of KACs. Third, urinary excretion is the main disposition pathway reported for Matr in humans [11] with possible involvement of renal clearance but the study design did not include the collection of urine samples, which would have provided the actual value of renal excretion rates of Matr in PBPK modeling. Additionally, the OCT transporter is proposed to involve in the renal clearance and the salivary excretion of Matr [28,35], which is needed to improve the physiological relevance of the models and will be investigated in future studies. Lastly, though the sample size (*n* = 8) for a single-dose PK study in healthy humans was optimum, the population pharmacokinetic estimation can further improve with a more representative US demographic.

## 5. Conclusions

In conclusion, our data provide convincing evidence to use saliva matrine levels as a patient compliance monitoring tool and verify our sampling approach for estimating population PK parameters of matrine for the ongoing chemoprevention clinical trials of ATB (NCT03459729; pharmacokinetics and NCT04278989; tumor inhibition) in oral cancer patients. The PBPK models of matrine adequately described the observed plasma concentration-time profiles of orally administered matrine in ATB in mice, rats, and humans. Therefore, the rodent data can be successfully used to predict human pharmacokinetics using the interspecies scaling approach. We will utilize this approach in the ongoing ATB multiple-dose pharmacokinetic study in cancer patients to validate the model for use in the efficacy of the clinical studies.

## Figures and Tables

**Figure 1 cancers-15-00089-f001:**
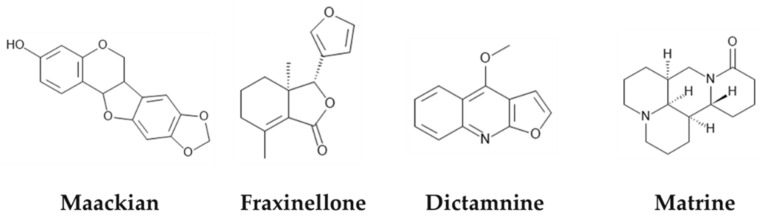
Chemical structure of ATB key active compounds.

**Figure 2 cancers-15-00089-f002:**
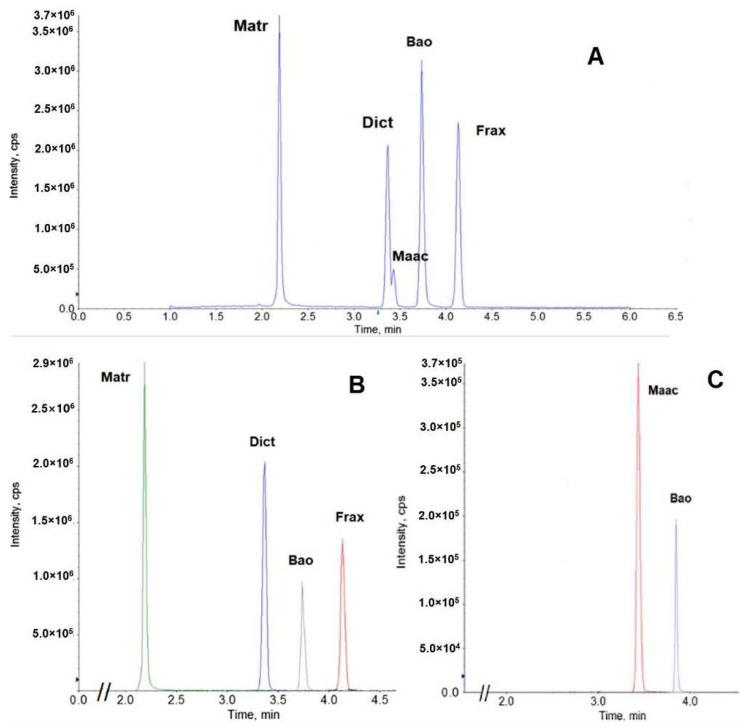
UPLC-MS/MS chromatogram of ATB key active compounds matrine (Matr), dictamnine (Dict), maackiain (Maac), fraxinellone (Frax) and the internal standard (Baohuoside I-Bao). (**A**) Total ion chromatogram, (**B**) chromatogram in positive mode, (**C**) chromatogram in negative mode.

**Figure 3 cancers-15-00089-f003:**
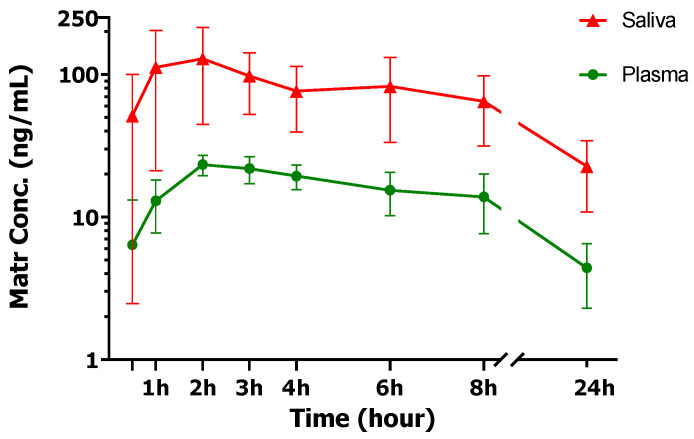
Plasma and saliva concentration-time profiles of Matr after single dose administration of ATB in healthy human volunteers (*n* = 8). Each data point represents the mean of individual plasma/saliva of 8 subjects with error bar (standard deviation).

**Figure 4 cancers-15-00089-f004:**
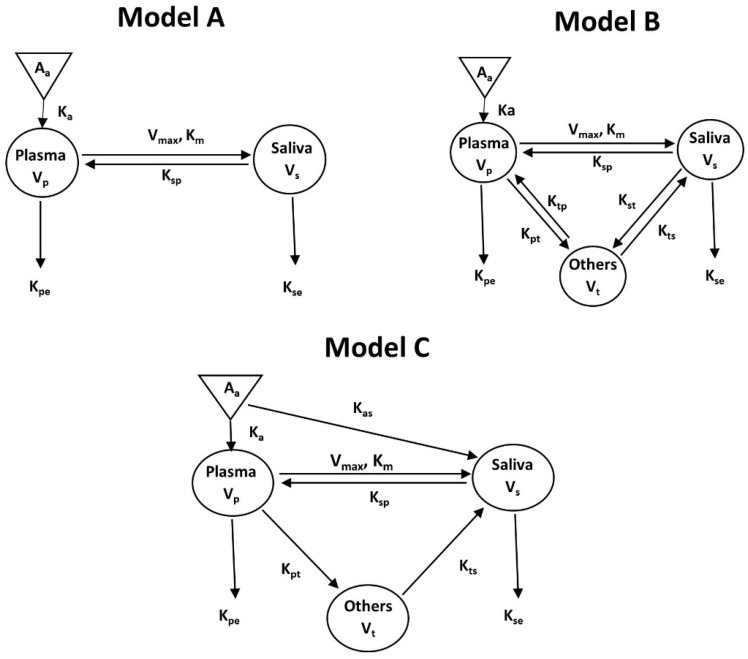
Proposed model structures for Matr in human plasma and saliva. The parameters in models (**A**–**C**) are described as V_p_, V_s_, V_t_: volume of distribution of the plasma, saliva, and other tissues compartment, respectively; CL_p_, CL_s_, apparent clearance of plasma and saliva, respectively; T_lag_: time of observation prior to the first observation with a measurable concentration; K_pt_, K_ts_, K_sp_: first-order rate constant between compartments; K_a_, K_as_: absorption rate constant to the central and salivary compartment, respectively; V_max_: the maximum velocity of drug transfer rate; K_m_: the substrate concentration at which the transfer rate is half its maximal value.

**Figure 5 cancers-15-00089-f005:**
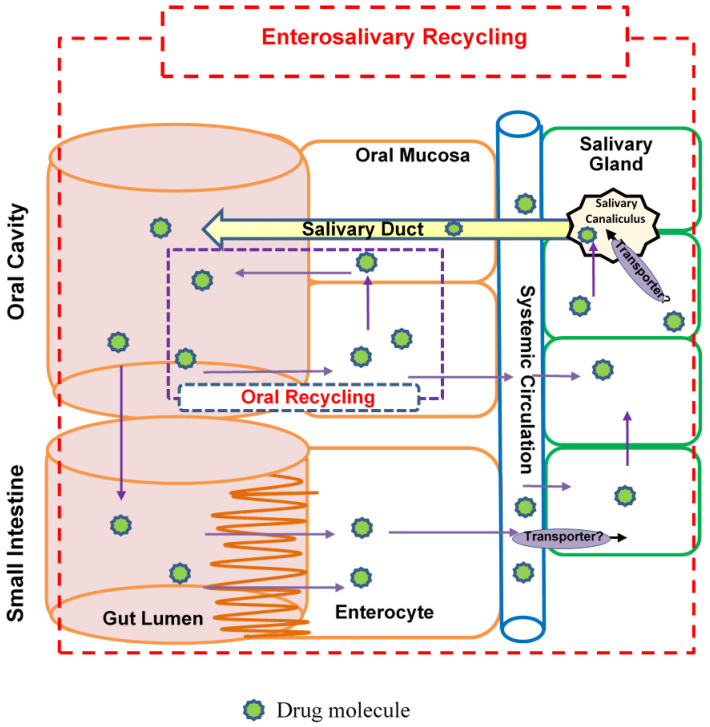
The proposed entero-salivary drug recycling scheme.

**Figure 6 cancers-15-00089-f006:**
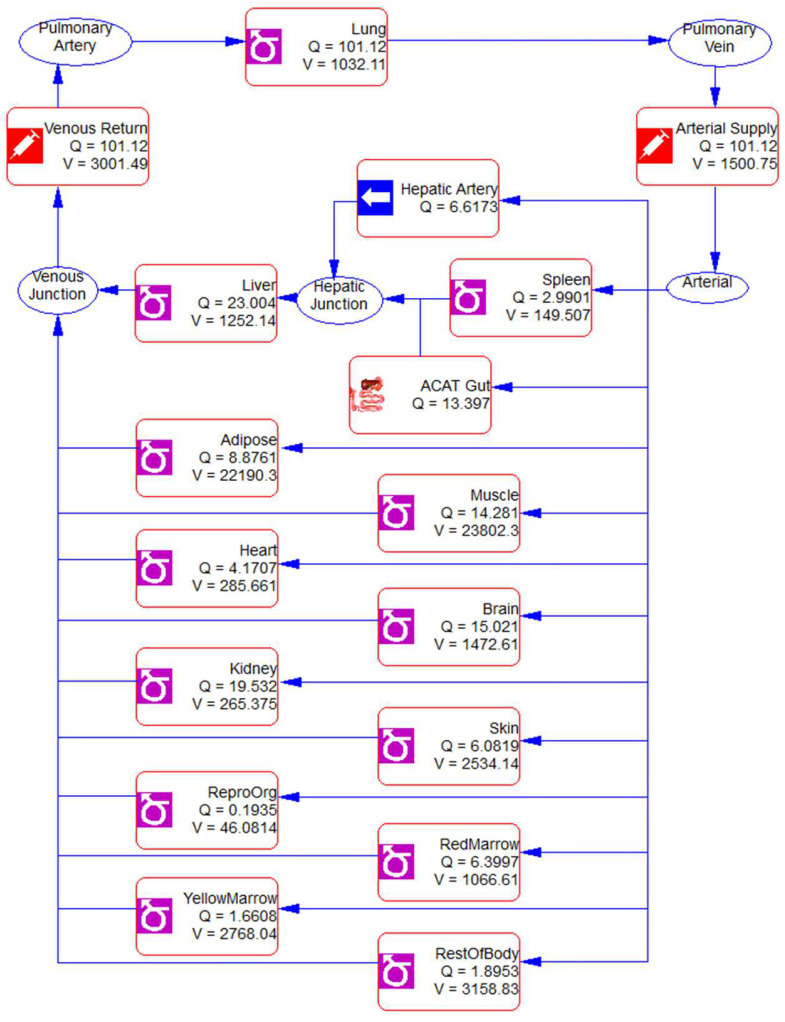
Model structure of the PBPK model of matrine using GastroPlus^®^ 9.8.

**Figure 7 cancers-15-00089-f007:**
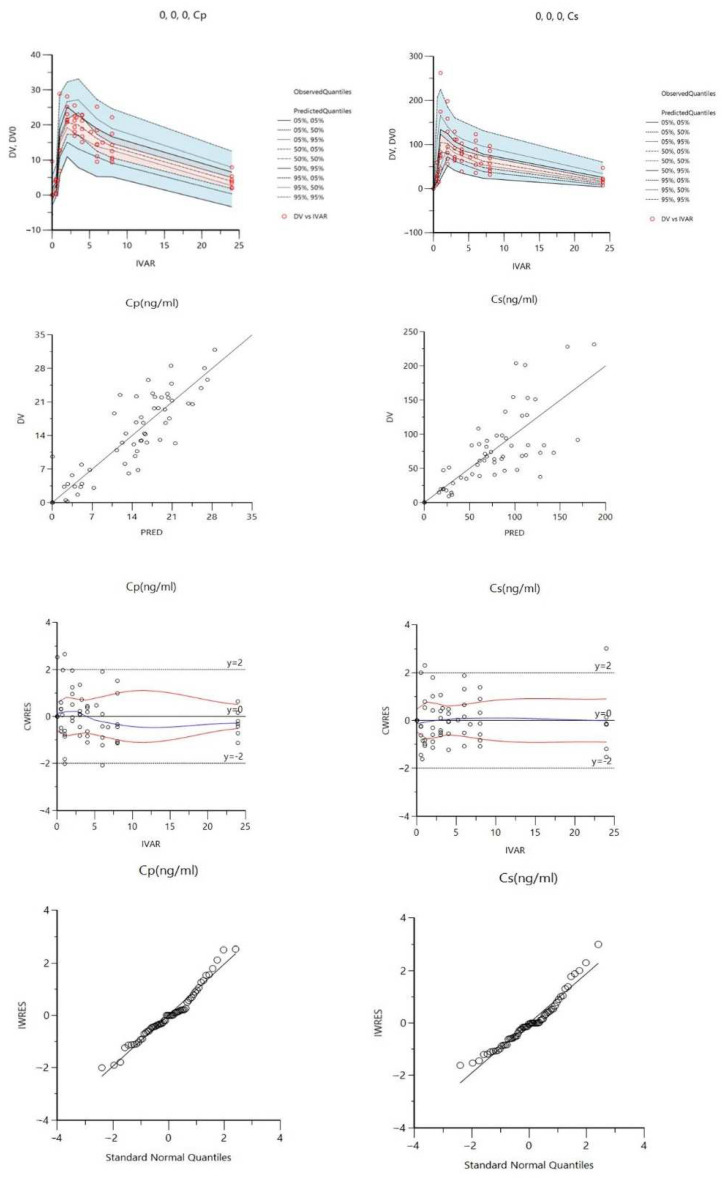
Best fit model result plots of Matr in human plasma and saliva from Model C (C_p_: plasma concentration, C_s_: saliva concentration, IVAR: independent variable—time, DV: dependent variable—blood concentration, IPRED: individual prediction, CWRES: conditional weighted residual).

**Figure 8 cancers-15-00089-f008:**
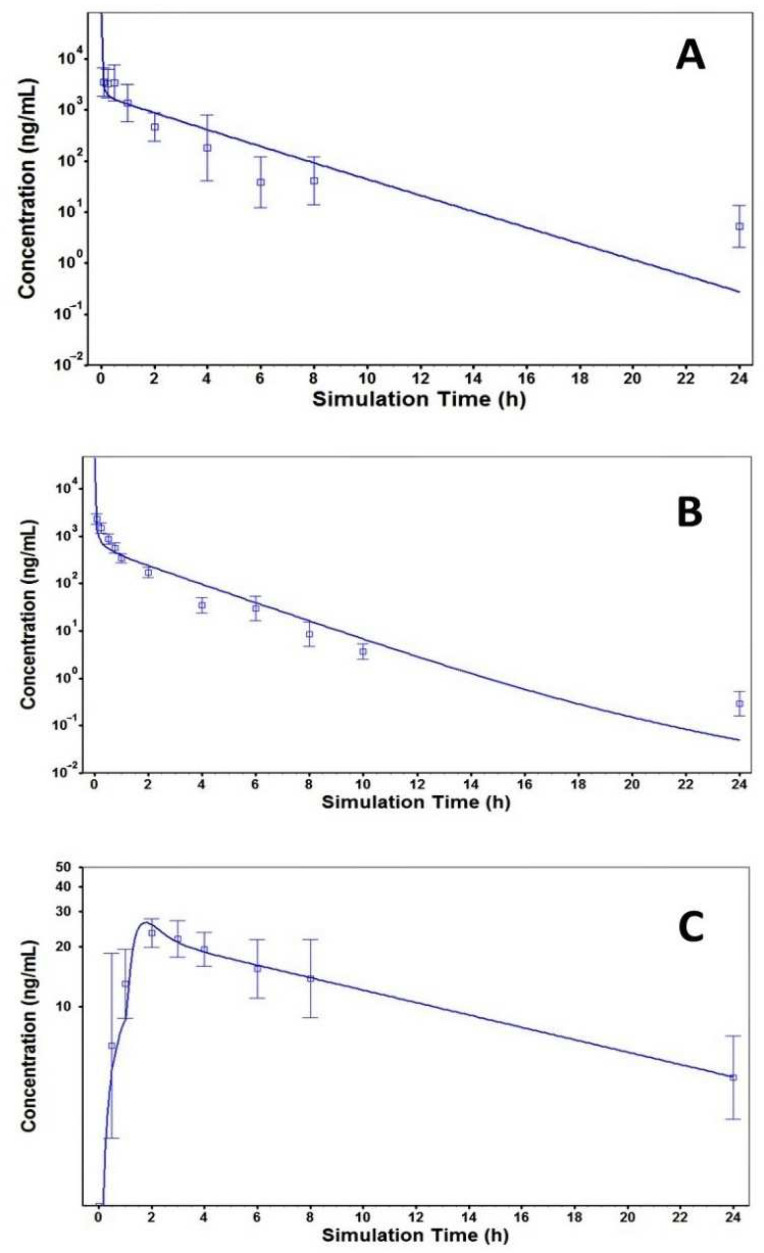
Simulated and observed plasma concentration results of Matr in mouse (2.8 mg/kg i.v., panel (**A**)), rat (2.0 mg/kg i.v., panel (**B**)) and human (2.6 mg p.o., panel (**C**)) with PBPK model. Observed data for mice were adopted from our recent publication [9], rat data were adopted from Yang et al. [15], and human data were generated in the current clinical study NCT04230057.

**Figure 9 cancers-15-00089-f009:**
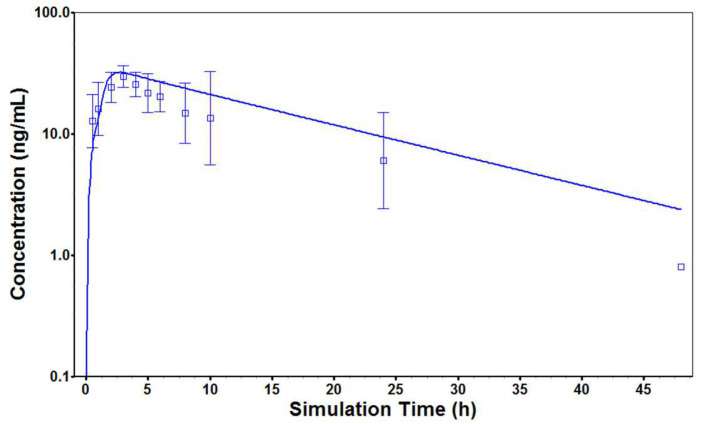
The predicted plasma profile of Matr in a 60 year-old American male generated based on our PBPK model (Figure 8C) was compared to the observed plasma profiles of *n* = 9 participants after a single dose of ATB (trade name Acapha^®^) tablets (equivalent to 4.8 mg Matr) in a PK study conducted at BC Cancer Agency, Canada. The study participants were former smokers previously diagnosed with bronchial dysplasia in the age group of 45–74 years. Additional demographics of the study participants were not mentioned in the thesis. Observed human PK data was adopted from the published doctoral thesis of Guanghua Gao at Simon Fraser University, British Columbia. [11].

**Table 1 cancers-15-00089-t001:** Botanical raw materials of ATB extract (percent weight).

Herbal Name		Chinese Name	Plants Parts Used	Form	% Content
Latin Name	Family				
*Sophora tonkinensis* Gagnep.	Fabaceae	Shan Dou Gen	Dried roots	Water extract	18–24
*Polygonum bistorta* L.	Polygonaceae	Quan Shen	Dried rhizome	Water extract	17–21
*Sonchus brachyotus* DC.	Asteraceae	Bei Bai Jiang	Dried whole plant	Water extract	17–23
*Prunella vulgaris* L.	Lamiaceae	Xia Ku Cao	Dried flower stem	Water extract	18–25
*Dioscorea bulbifera* L.	Dioscoreaceae	Huang Yao Zi	Dried rhizome	Water extract	3–6
*Dictamnus dasycarpus* Turcz.	Rutaceae	Bai Xian Pi	Dried root bark	Powder	8–12

**Table 2 cancers-15-00089-t002:** Amounts of ATB key active compounds in the ATB supplement tablets.

Analyte	Matrine	Maackiain	Dictamnine	Fraxinellone
Amount (μg/g)	988.7 ± 170.2	15.5 ± 0.6	11.7 ± 2.3	21.2 ± 3.8
Amount (μg/8 tablets)	2610.2 ± 449.3	40.9 ± 1.6	30.9 ± 6.1	56.0 ± 10.0

**Table 3 cancers-15-00089-t003:** List of final parameters of matrine that were used to implement the PBPK models.

Parameter	Value	Resources
Molecular weight (g/mol)	248.36	
Log D	1.45	ADMET Predictor^®^ 10.2
pKa	9.49	ADMET Predictor^®^ 10.2
Solubility at pH 7.4 (mg/mL)	50.00	ApexBio website [16]
Rbp	0.97	Gao et al. [11]
Fup (%)	95.85	ADMET Predictor^®^ 10.2
First pass effect (%)	26.00	Bui et al. [9]
P_eff_ (Caco-2, cm/s)	4.25 × 10^−5^	Yang et al. [15]

**Table 4 cancers-15-00089-t004:** Demographic summary of all participants (*n* = 8).

	Mean (±SD)	Median (Range)
Age, Years	32.88 (4.02)	34 (26–27)
Height (cm)	164.69 (8.10)	166 (151–176)
Weight (lbs)	151.4 (32.39)	148.5 (103–192.2)
Body Mass Index (kg/m^2^)	25.18 (4.35)	25.2 (20.5–32)

**Table 5 cancers-15-00089-t005:** Pharmacokinetic parameters of matrine in human plasma and saliva.

Parameter	T_max_ (h)		Half-Life (h)		C_max_ (ng/mL)			AUC_0–24_ (ng * h/mL)		
Subject	Saliva	Plasma	Saliva	Plasma	Saliva	Plasma	Saliva/Plasma Ratio	Saliva	Plasma	Saliva/Plasma Ratio
1F	1	2	7.59	15.12	91.61	28.06	3.26	845.11	295.05	2.86
2F	2	2	26.39	17.06	120.81	23.60	5.12	1067.94	228.16	4.68
4M	2	2	18.98	10.66	201.34	28.56	7.05	1884.98	408.92	4.61
5M	3	2	11.13	9.84	157.71	21.28	7.41	1024.67	247.39	4.14
8F	1	3.3	5.42	6.29	231.60	31.91	7.27	2064.14	428.83	4.81
9M	1	1	12.23	8.65	204.07	22.48	9.08	911.28	231.84	3.93
11M	0.5	3	7.38	6.99	153.18	22.77	6.73	1487.35	173.50	8.57
12M	2	2	9.83	8.43	127.11	22.70	5.60	1133.48	240.73	4.71
AVG ± SD	1.56 ± 0.82	2.16 ± 0.69	12.37 ± 7.01	10.38 ± 3.83	152.98 ± 53.54	25.17 ± 3.82 *	6.4 ± 1.8	1302.37 ± 459.28 *	278.34 ± 84.90 *	4.8 ± 1.7

AUC: area under the plasma/saliva concentration-time curve; C_max:_ maximum concentration of matrine in plasma/saliva; T_max_: time to achieve maximum concentration of matrine in plasma/saliva. (*) indicates the significant difference between plasma and saliva (*p* < 0.05).

**Table 6 cancers-15-00089-t006:** List of the final PK parameters for population co-modeling of plasma and saliva concentration of matrine in human (Model C).

Parameter	Estimate	CV (%)	Units
V_p_	1.42 ± 0.08	5.32	L/kg
CL_p_	0.02 ± 0.01	48.57	L/h/kg
T_lag_	0.44 ± 0.01	2.55	h
K_a_	1.66 ± 0.27	16.29	1/h
V_s_	0.02 ± 0.01	35.98	L/kg
V_max_	11.88 ± 1.27	10.66	mg/h/kg
K_m_	279.52 ± 5.73	2.05	mg/L
V_t_	0.62 ± 0.04	7.03	L/kg
K_pt_	0.11 ± 0.02	21.34	1/h
CL_s_	0.05 ± 0.01	17.55	L/h/kg
K_ts_	23.42 ± 2.33	9.93	1/h
K_sp_	1.25 ± 0.28	22.69	1/h
K_as_	0.39 ± 0.14	36.28	1/h

V_p_, V_s_, V_t_: volume of distribution of the plasma, saliva, and other tissues compartment, respectively; CL_p_, CL_s_, apparent clearance of plasma and saliva, respectively; T_lag_: time of observation before the first observation with a measurable concentration; K_pt_, K_ts_, K_sp_: first-order rate constant between compartments; K_a_, K_as_: absorption rate constant to the central and salivary compartment, respectively; V_max_: the maximum velocity of drug transfer rate; K_m_: the substrate concentration at which the transfer rate is half its maximal value.

**Table 7 cancers-15-00089-t007:** Simulated and observed pharmacokinetic parameters for matrine in mice, rats, healthy volunteers and human subjects in the PBPK model.

Parameter	Mice		Rats		Healthy Volunteers		Human Subjects	
	Observed	Predicted	Observed	Predicted	Observed	Predicted	Observed	Predicted
C_max_ (ng/mL)	NA	NA	NA	NA	25.2	26.3	29.7	32.1
T_max_ (h)	NA	NA	NA	NA	2.2	1.8	3.0	2.7
AUC_0-T_ (μg * h/L)	5131.1	5448.7	1717.6	1718.1	278.3	270.8	412.2	570.6

NA: Not applicable for i.v. data in mice and rats.

## Data Availability

The data that support the findings of this study are available from the corresponding authors, Rashim Singh and Ming Hu, upon reasonable request.

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
