# Peer review of "Pharmacokinetic Basis for Using Saliva Matrine Concentrations as a Clinical Compliance Monitoring in Antitumor B Chemoprevention Trials in Humans"

_cancers, 2022, doi:10.3390/cancers15010089_

Round 1

Reviewer 1 Report (Previous Reviewer 2)

The paper shows an interesting study and as I can see in the presented manuscript the flaws present in the previous version are already corrected. I would like only to ask for the statement form, line 127: To mimic the future clinical trial in patients, no further standardization of saliva samples was made in the current study". Honestly speaking I do understand this. Could the Authors explain why they omitted the standardization of saliva samples?

Author Response

In real-world scenario, chemoprevention trials are usually long-term clinical trials, where sample collection happens mostly at home by the patients themselves. For the purpose of reducing variability in the sampling time and volume of saliva sample collection, we plan to instruct patients to stop drinking or eating 15 minutes before sampling and spit into the saliva sample kit for salivary collection, However, we expect high variability in saliva samples collection due to the limitations in patient compliance to the procedure/instructions over a long period of time. Therefore, in the current study, though we did standardize the collection method, we choose to not further standardize the samples to have some variability as expected in the real chemoprevention clinical trials.  Another alternative is to measure the patient’s creatine levels in the saliva and use that to normalize the results. We will investigate in future studies.

Reviewer 2 Report (Previous Reviewer 4)

The authors have covered the all points for the revision and significantly improved. It can be considered for publication.

Author Response

Thank you for reviewing the manuscript. No other question.

This manuscript is a resubmission of an earlier submission. The following is a list of the peer review reports and author responses from that submission.

Round 1

Reviewer 1 Report

Bui and coworkers has described the effect of ATB tablets intake on the level of one of its active ingredient - matrine in saliva and blood. The pharmacokinetic parameters were evaluated and compared (blood versus saliva) what led authors to conclude that the measurement of marine in saliva can be a biomarker of secretion of ATB from plasma to saliva. Additionally the PBPK modelling was performed to compare the effect in mice rats and humans. The study is well described however there are some issue need to be clarified prior publication, in particular both above cited conclusions should be rediscussed and reedited, see below.

MAJOR:

Discussion – lines 291-297: these are too far going conclusions.  Lines 201-294: The authors provided the results obtained only for one component, thus the conclusion may be made only for this 1 element. Similarly lines 355-359 – since the authors did not deliver the results of the remaining components the conclusion can be made for the matrine only. Please refer to the above.

The authors state that they compare the simulated results and observed data originating form mice, rats, and human. Please provide the source of observed data you use for comparison with your model.

What is a difference between Fig7C and Fig8. Which data originate from the ClinicalTrial. Was the “American male” (The Caucasian should be used instead American) one of the study participant? In Table 8 the authors write Caucasian males – was there more than one man included in the study. Please explain also the number n=9 in the captions of Fig. 8, and how it refers to “American male”. The above issues should be clarified.

Please include the description of ATB active indgridients: Matr Dict Maac Frax in the introduction and include shortcuts after first use.

Provide the reference to FDA guidance (line 88) and PhD thesis (line 219)

Please provide the information how many participants were in total and why only 8 was included in the model, please refer to /doi.org/10.5334/joc.72 to discuss if 8 participants ensure properly conducted statistical analysis.

MINOR:

-              Table 1, Table 2 and text – please decide if use Antitumor B or ATB shortcut.

-              In some part of text the font size is different (e.g. 2.1, 3.3)

-              please correct captions in Fig.1

-              explain what RSD stands for (2.6, line 161)

-              editorial error – line 212 (Fig.3…)

-              please review the text to correct some stylistic errors,  e.g line 45-46, 269-270

Reviewer 2 Report

In principle, the paper is well prepared and presented. I would like to rise two issues.

- Did the authors somehow standardise the samples of saliva? Although the blood has a very stable composition the saliva is produced in different amounts in different persons therefore comparison of the concentrations of tested drugs in saliva may be misleading.

-the paper contains a lot of not explained abbreviations in the text but also in the figures e.g. Fig 5 table 5. They are not so commonly used as AUC, Tmax or Cmax.  They must be explained otherwise it is impossible to understand the paper.

Reviewer 3 Report

In this study the authors reported a clinical outcome modeling regarding the PK in plasma and saliva. Through oral administration, the ATB exhibits dual PK profiles which was further analyzed through a population model and a PBPK model, and the tracer of e.g. Matr could serve as a competitive biomarker that enable further development of the salivary secretion based PKPD analysis. The manuscript follows a good workflow and is comprehensive for a general understanding of the drug and the concept of saliva PK. On top of that I would like to request the authors to provide more detailed description of the PBPK model, including the chart of design, the intermediate variables and the reference sources. Other than that it would be beneficial to include the model files for validation and sensitivity analysis purposes. Therefore, I would suggest a minor revision of the manuscript, and some minor issues are listed below:

1.       Please revise and include the diagram where you showed the compartmental models for both popPK and PBPK, including the intermediate parameters in table.

2.       Revise Fig. 6, replot with confidence interval whenever applicable, and, in supplemental if preferred, please add an extra plot of Visual Prediction Check, a plot with potent correlation of covariates/model parameters, show the distribution of random effects.

3.       Discuss extensively the physiological meaning of the PK parameters obtained.

4.       Based on the models you obtained, is there any predictive work you can do regarding the efficacy, or directing the clinical development in the future?

Reviewer 4 Report

Authors have written a research article titled “Pharmacokinetic Basis for Using Saliva Matrine Concentrations as a Clinical Compliance Monitoring in Antitumor B Chemoprevention Trials in Humans”. The manuscript has been well written and well designed. However, some sentence framing, spelling error and grammar of the manuscript must be corrected with consideration following points.

1.      In the introduction section, the second line corrects “power” as powder.

2.      Page no 2, line no 80, the degree symbol is inconsistent; please correct it.

3.      Line no 105, please correct the grammar of the sentence.

4.      The presented SD in table 3, whether it represents +SD or –SD or both, please mention.

5.      Line no 212, “Fig 3Error! Reference source not found” what dose indicates?